# Supramolecular tessellations by the exo-wall interactions of pagoda[4]arene

Xiao-Ni Han[1,2], Ying Han[1] & Chuan-Feng Chen [1,2] ✉

Supramolecular tessellation has gained increasing interest in supramolecular chemistry for its structural aesthetics and potential applications in optics, magnetics and catalysis. In this work, a new kind of supramolecular tessellations (STs) have been fabricated by the exo-wall interactions of pagoda[4]arene (P4). ST with rhombic tiling pattern was first constructed by P4 itself through favorable $\pi\cdots\pi$ interactions between anthracene units of adjacent P4. Notably, various highly ordered STs with different tiling patterns have been fabricated based on exo-wall charge transfer interactions between electron-rich P4 and electron-deficient guests including 1,4-dinitrobenzene, terephthalonitrile and tetrafluoroterephthalonitrile. Interestingly, solvent modulation and guest selection played a crucial role in controlling the molecular arrangements in the co-crystal superstructures. This work not only proves that P4 is an excellent macrocyclic building block for the fabrication of various STs, but also provides a new perspective and opportunity for the design and construction of supramolecular two-dimensional organic materials.

[1] Beijing National Laboratory for Molecular Sciences, CAS Key Laboratory of Molecular Recognition and Function, Institute of Chemistry, Chinese Academy of Sciences, 100190 Beijing, China. [2] University of Chinese Academy of Sciences, 100049 Beijing, China. ✉email: cchen@iccas.ac.cn

Tessellation or tiling is a process of creating periodic patterns by arranging one or more polygons to cover a plane entirely without overlaps or gaps, which has been used to tile kinds of polygons for decoration in art since antiquity[1–3]. Moreover, tessellation not only shows structural esthetics for appreciation but also has wide applications in numerous fields, such as mathematics, art, physics, biology, architecture, and molecular science[4–8]. The design of tessellations at molecular level has gained increasing interest in the past decades for the potential applications in optics[9], magnetics[10], and catalysis[11]. Molecular tiling, such as self-assembly of DNA molecules[12–14], polymeric systems[15], quasicrystals[16], liquid crystal engineering[17], and two-dimensional (2D) covalent organic frameworks[18–20] (COFs), have been reported in recent years. Especially supramolecular chemistry provides fascinating routes to achieve versatile tessellations at molecular level by making use of the favorable noncovalent interactions[21–26].

In most of the supramolecular tessellation (tessellation at molecular level by making use of the favorable noncovalent interactions) systems, the design of vertices is a universal way to fabricate various tessellations since the symmetry element of vertex could determine the geometry superstructures of molecular tessellation[27–30]. In comparison, construction of the tessellations with polygonal macrocycles offers an alternative and effective method[31–36]. However, this method is less explored due to the lack of perfect polygonal macrocycles and the difficulty in how to achieve large-scale controllable self-assembly using the macrocycles as building blocks. Therefore, searching for favorable polygonal macrocycles to achieve the fantastic tiling of supramolecular tessellations is still a big challenge and highly desirable.

Three regular tessellations are shown in Fig. 1, each of them has one type of high-symmetry vertex as highlighted in orange and the basic building blocks are triangle, hexagon, and square, respectively. Stoddart's group[35] first reported the supramolecular tessellations by a rigid naphthalene diimide triangle (Fig. 1a). Recently, Huang et al.[36] constructed a new kind of supramolecular tessellations via exo-wall interactions of pillar[6] arene with hexagonal configuration (Fig. 1b). However, no perfect tiling based on square macrocyclic building blocks through exo-wall donor–acceptor interactions have been fabricated so far due to the lack of such macrocycles and the difficulty in utilizing favorable noncovalent interactions properly. Recently, we[37] reported a new type of macrocyclic arene, pagoda[4]arene (P4). It was composed of four identical anthracene units to form a square configuration and showed intriguing host–guest properties, which provided us the opportunity to explore the square supramolecular tessellations with P4 as a building block.

In this work, we report a new kind of supramolecular tessellations at the molecular level, which were fabricated by the exo-wall interactions of P4 (Fig. 1c). First, the supramolecular tessellation with rhombic tiling pattern of a shape-adjustable P4 polygon was obtained through favorable π···π interactions between the anthracene units of adjacent P4. Solvent modulation could induce intriguing changes in both the shapes and the arrangements of P4 in the resulting crystal superstructures. Notably, various highly ordered supramolecular tessellations with different tiling patterns were fabricated by the exo-wall interactions between electron-rich P4 and electron-deficient guests, including 1,4-dinitrobenzene (DNB), terephthalonitrile (TPN), and tetrafluoroterephthalonitrile (TFTN) (Fig. 2). Interestingly, it was further found that both solvent modulation and guest selection played a vital role in controlling the molecular arrangements of the resulting crystal superstructures. Moreover, the X-ray crystallography analysis was used to characterize the superstructures and provided valuable information of the intermolecular interactions that could effectively influence the molecular arrangements in the solid state.

## Results

### Design of P4-based supramolecular tessellation through the exo-wall interactions
It is necessary to introduce effective driving forces to achieve highly ordered 2D supramolecular

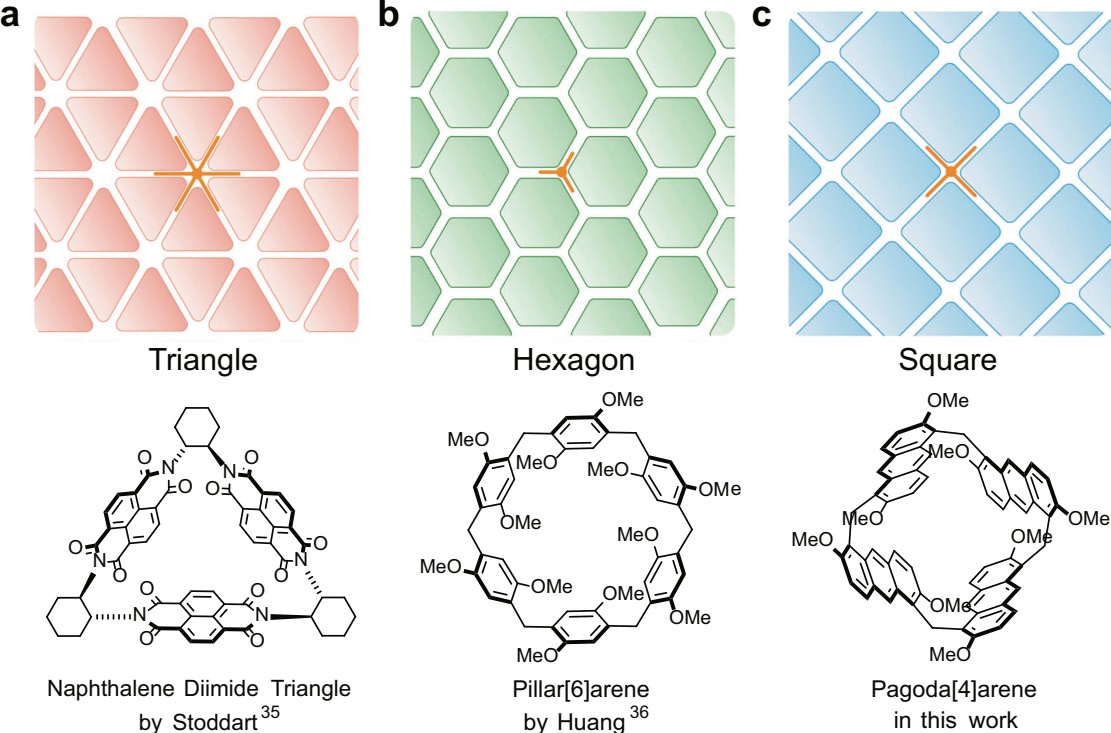

**Fig. 1 Three regular supramolecular tessellation building blocks. a** Naphthalene diimide triangle. **b** Pillar[6]arene hexagon. **c** Pagoda[4]arene square.

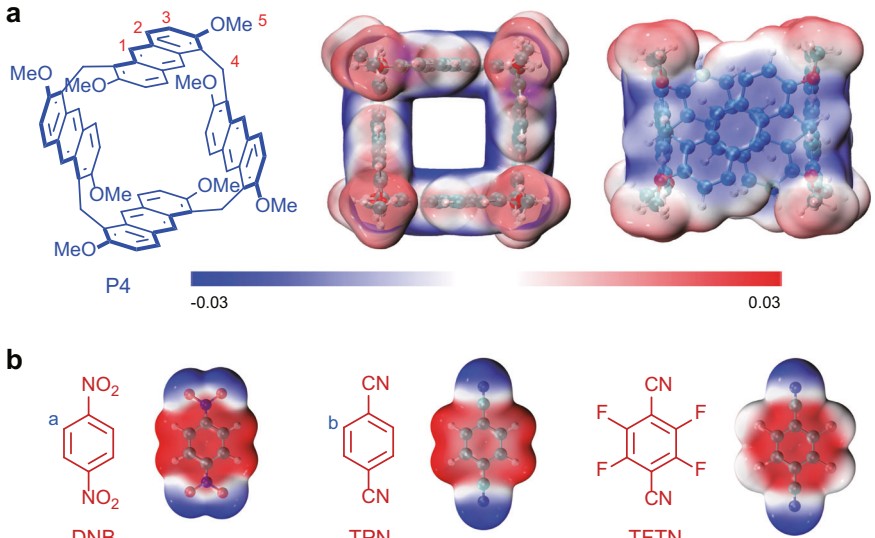

**Fig. 2 Chemical structures and electrostatic potential maps.** Chemical structures and electrostatic potential maps of **a** P4 and **b** guests DNB, TPN, and TFTN as building blocks for the design of supramolecular tessellations.

tessellations by macrocyclic polygons, and exo-wall bindings such as π···π and charge-transfer (CT) interactions could provide feasible approaches[38–46]. In macrocyclic chemistry, exo-binding was rarely explored[47–53] compared with the abundant endo-cavity binding studies[54–61]. P4 is composed of four identical electron-rich anthracene subunits and shows $D_4$ symmetrical polygon structure. The inspiration of P4-based supramolecular tessellation was to utilize the favorable π···π interactions between the anthracene subunits of adjacent P4 and introduce CT interactions between electron-rich P4 and the electron-deficient planar guests in the building blocks. Propagation of the exo-wall interactions orderly was supposed to form highly uniform supramolecular tessellations at the molecular level.

First, the electrostatic potential (ESP) maps of P4 and related guests were calculated to analyze the possibility of the external host–guest complexation[62]. As shown in Fig. 2a, the ESP map of P4 showed that both sides (in and out) of the anthracene walls of regular polygon P4 were highly electronegative. Besides, the aromatic area of DNB, TPN, and TFTN displayed strong electropositivity (Fig. 2b), which was prone to forming CT interactions with the electronegative walls of P4. Therefore, it is reasonably speculated that DNB, TPN, and TFTN could probably work as the linkers to form "exo-wall" interactions with P4 polygon in proper conditions. Therefore, the regular structure of electron-rich P4 offered an excellent opportunity to construct highly ordered supramolecular tessellations through π···π interactions between adjacent P4 and CT interactions between P4 and the electron-deficient guests.

**Investigation of CT interactions in solution.** Initially, the interactions of P4 with DNB, TPN and TFTN were investigated in solution with nuclear magnetic resonance (NMR) spectroscopy. We first studied the interaction between P4 and DNB. In the [1]H NMR spectrum of a mixture of P4 and DNB in CDCl₃, the peak of $H_a$ on DNB showed broaden upfield shift compared with that of free DNB. Meanwhile, peaks of $H_1$, $H_2$, and $H_4$ on P4 also exhibited upfield shifts (Supplementary Fig. 7). Moreover, we also studied the interactions of P4 and DNB in CD₂Cl₂. It was found that, in the spectrum of the mixture, the peaks of $H_a$ on DNB and $H_1$ on the anthracene ring also showed chemical shift changes (Supplementary Fig. 8). These results confirmed the interactions between P4 and DNB. Similarly, the complexations between P4

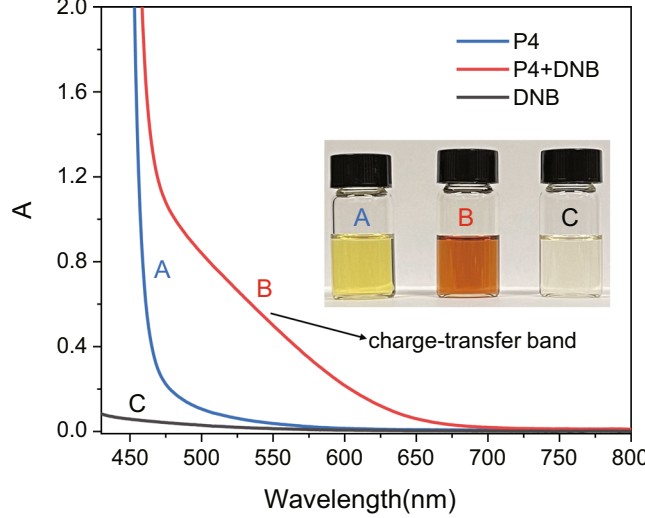

**Fig. 3 UV-vis spectra.** UV-vis spectra (CHCl₃): (A) P4 (3.0 mM); (B) P4 (3.0 mM) and DNB (6.0 mM); (C) DNB (6.0 mM). The inserted optical image shows the solution color change because of the charge-transfer interaction between P4 and DNB.

with TPN and TFTN were also confirmed by [1]H NMR spectra in both of CDCl₃ and CD₂Cl₂ (Supplementary Figs. 9–12). Besides, the [19]F NMR signal of TFTN in a mixture of TFTN and P4 shifted upfield compared with that of free TFTN (Supplementary Figs. 13 and 14), which further confirmed the interactions between P4 and TFTN.

Notably, we found that, when the yellow solution of P4 (3.0 mM) and colorless DNB (6.0 mM) solution were mixed in CHCl₃, a significant color change occurred and an orange solution was observed (inset picture in Fig. 3). We further investigated the CT interactions through ultraviolet–visible (UV-vis) spectroscopy. Spectrum B in Fig. 3 exhibited an obvious broad absorption band at 460–700 nm that was different from the spectrum of either P4 (A) or DNB (C), indicating a characteristic absorption band of the CT complexation between P4 and DNB. Moreover, although P4 and TPN did not show obvious CT interaction in solution (Supplementary Fig. 24), strong CT

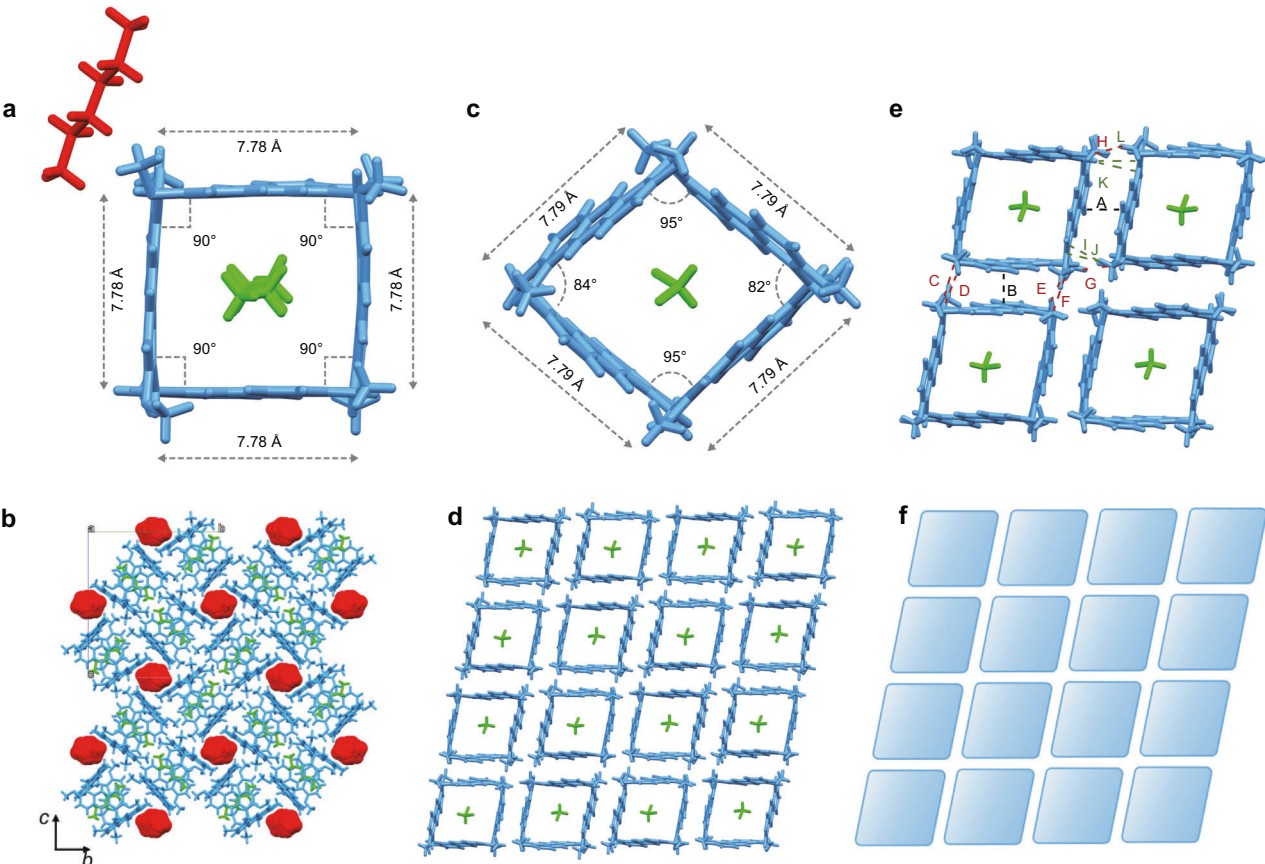

**Fig. 4 Supramolecular tessellations of P4 through π···π interactions. a** Crystal structure of *n*-hexane@P4 showing the asymmetric unit. **b** Packing mode of *n*-hexane@P4 along *a* axis. **c** Crystal structure of CH₂Cl₂@P4 showing the asymmetric unit. **d** Two-dimensional regular rhombic tiling in a plane. **e** Interactions between adjacent P4 in the rhombic basic unit. **f** Schematic representation of the rhombic tiling. Different colors represent the symmetry equivalence.

interaction with P4 could be observed when the hydrogen atoms of TPN were replaced with fluorine atoms as TFTN (Supplementary Fig. 25).

**Supramolecular tessellations of P4 through π···π interactions.** The synthesis of P4 was carried out according to the reported literature procedure[33]. Although the crystal structures of *n*-hexane@P4 and CH₂Cl₂@P4 have been already reported by our group before[33], the packing modes of them have not been thoroughly studied. Herein, we studied the superstructures of these crystals and analyzed the different forces that could enhance or prevent the formation of highly ordered supramolecular tessellations.

As shown in Fig. 4a, there existed one macrocycle P4 and two *n*-hexane molecules in the asymmetric unit of *n*-hexane@P4. Among them, one *n*-hexane molecule was encapsulated in the cavity of P4 and the other *n*-hexane was in the outside of the cavity. P4 adopted a symmetrical square structure with the side lengths of 7.78 Å and the angles between the four adjacent anthracene planes were all 90°. However, the *n*-hexane molecules outside the cavities of P4 were arranged in a disorderly way (Fig. 4b), which prevent P4 units to form favorable π···π interactions with adjacent P4 in a face-to-face fashion. As a result, the polygon P4 of *n*-hexane@P4 formed neither tubular superstructure nor a tilling pattern in the packing mode (Supplementary Fig. 26).

It was interesting to find that macrocyclic polygon P4 could show adjustable cavity shapes with different solvent molecules inside its cavity. For the crystal of CH₂Cl₂@P4, there existed one

macrocyclic polygon P4 and one CH₂Cl₂ molecule inside its cavity in the asymmetric unit (Fig. 4c). The macrocyclic polygon P4 adopted a rhombus structure with the side lengths of 7.79 Å and the angles between the four adjacent anthracene planes of 95°, 84°, 95°, and 82°, respectively. Remarkably, tessellation with rhombic tiling pattern could be formed by P4 itself mainly through π···π interactions between adjacent P4 (Fig. 4d). As shown in Fig. 4e, all the polygon faces of P4 could interact through perfect parallel face-to-face π···π stacking interactions with adjacent P4 with plane–plane distances of 3.59 and 3.53 Å (A and B), and the corresponding dihedral angles of the adjacent anthracene planes were all 0°. In addition, there existed multiple C-H···O interactions (C, D, E, F, G, and H) with the distances of 2.70, 2.50, 2.50, 2.70, 2.57, and 2.57 Å, respectively, between the oxygen atom of P4 and methoxy hydrogen atoms of adjacent P4. Besides, C-H···π interactions (I, J, K, and L) with the distances of 2.62, 2.68, 2.68, and 2.62 Å between adjacent P4 were also observed. All these interactions worked together to form the stable supramolecular arrangement and propagated in a plane to form a layer-like uniform rhombic superstructure (four rhombus around a vertex) with "edge-to-edge" tiling manner as shown in Fig. 4d–f. It was further found that the adjacent P4 were enantiomers with each other in the tiling plane (Supplementary Fig. 27). This interesting phenomenon of alternately arranged enantiomers could be ascribed to the parallel face-to-face stacking interaction between two anthracene units of enantiomers of P4, which is more favorable than interacting with P4 with same configuration by two anthracene units stacked in off-set face-to-face manner (Supplementary Fig. 28) for the uniform

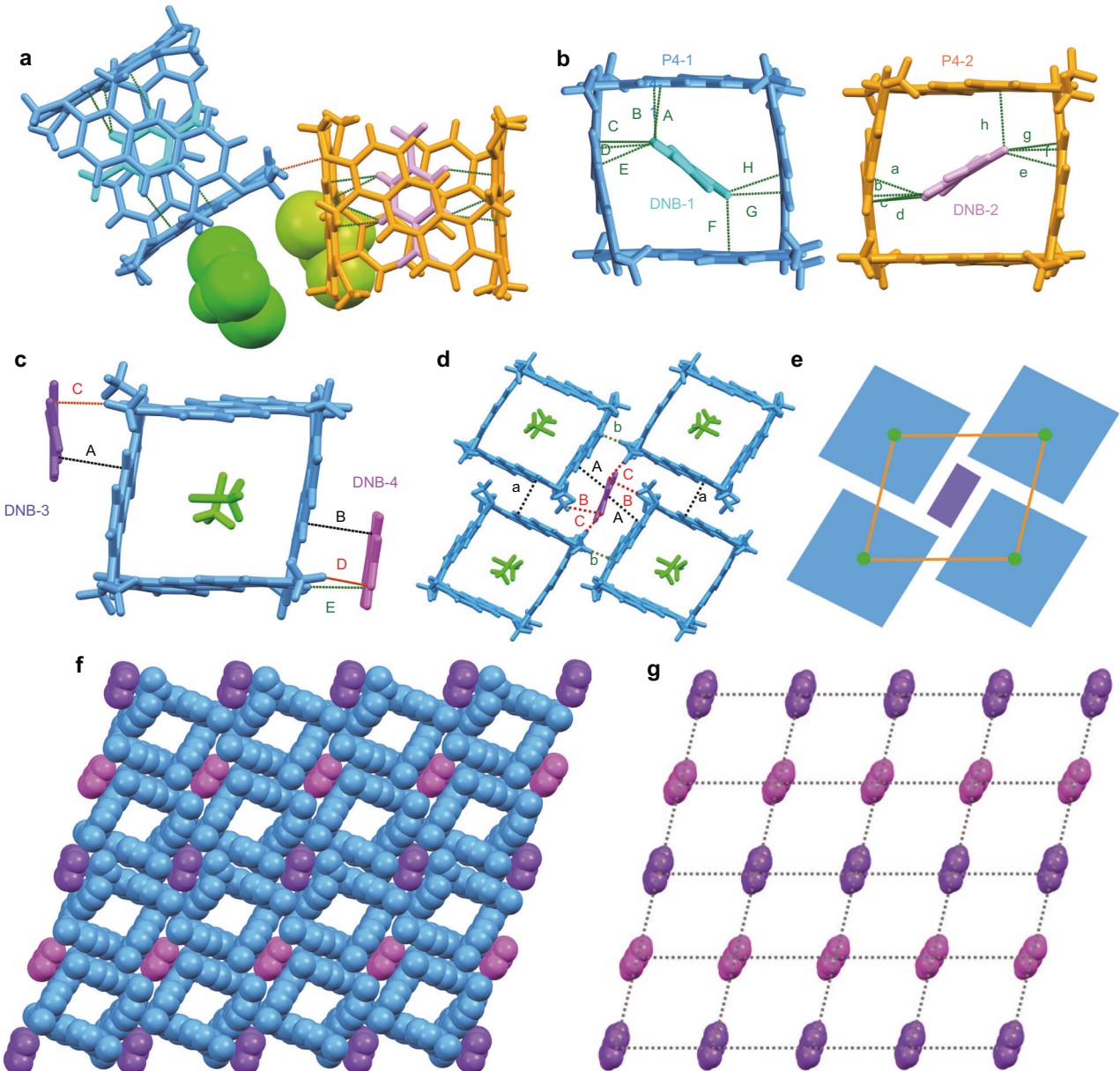

**Fig. 5 Supramolecular tessellations of P4 with DNB.** Crystal structure of P4-DNB$_a$ showing **a** side view of the asymmetric unit; **b** the interactions between P4 and DNB. Crystal structure of P4-DNB$_b$ showing **c** the asymmetric unit; **d** the basic tiling unit; **e** schematic representation of the basic tiling unit; **f** two-dimensional tiling of the uniform superstructure formed by P4 and DNB in a plane; **g** regular parallelogram grids formed by DNB molecules. Different colors represent the symmetry equivalence. Green, black, and red dotted lines represent the noncovalent interactions. Crystallographically independent P4 and DNB molecules were marked as P4-1, P4-2, DNB-1, DNB-2, DNB-3, and DNB-4, respectively.

arrangement. Moreover, although tubular structures were not observed in this crystal superstructure, adjacent P4 units could stack to form infinite one-dimensional supramolecular channels along *c* axis by a set of CH···π interactions in the stacking directions (Supplementary Fig. 29). Surprisingly, enantiomers of P4 could also be arranged alternately along *c* axis (Supplementary Fig. 30).

**Supramolecular tessellations of P4 with electron-deficient guests through CT interactions.** Relevant CT assemblies as co-crystals based on electron-rich P4 and electron-deficient guests have been prepared under different crystallization conditions. Interestingly, solvent modulation and guest selection played a vital role in controlling the molecular arrangement in the resulting crystal superstructures. Co-crystallization of P4 with

DNB, TPN, and TFTN in different crystallization conditions resulted in seven sets of co-crystals as P4-DNB$_a$, P4-DNB$_b$, P4-TPN$_a$, P4-TPN$_b$, P4-TFTN$_a$, P4-TFTN$_b$, and P4-TFTN$_c$, and the crystallographic data for all the co-crystals are summarized in Supplementary Tables 2–4.

**Supramolecular tessellations of P4 with DNB.** P4-DNB$_a$ crystallized in the monoclinic $P2_1/n$ space group with two P4, two DNB, and two CHCl$_3$ molecules in the asymmetric unit (Fig. 5a and Supplementary Figs. 32 and 33) by slow diffusion of methanol to a solution of P4 and DNB in CHCl$_3$. Among them, each DNB molecule was encapsulated in the cavity of P4, respectively, and there existed C-H···O interaction with distance of 2.66 Å between two P4. Besides, DNB molecules inside the cavity were in close contact with P4. As shown in Fig. 5b, there

existed multiple C-H⋯π interactions between P4-1 and encapsulated DNB-1 with the distances of 2.85 (A), 2.87 (B), 2.87 (C), 2.76 (D), 2.83 (E), 2.84 (F), 2.83 (G), and 2.75 (H) Å, respectively. And there also existed multiple C-H⋯π interactions between P4-2 and encapsulated DNB-2 with the distances of 2.69 (a), 2.89 (b), 2.78 (c), 2.83 (d), 2.87 (e), 2.79 (f), 2.68 (g), and 2.82 (h) Å, respectively. It was found that when DNB molecules were encapsulated in the cavities of P4 to form traditional in-cavity host–guest complexation, solvent CHCl$_3$ molecules in the outside of the cavities prevented the uniform molecular arrangement. As a result, the packing of P4 and DNB formed neither tubular superstructure nor a tiling pattern (Supplementary Fig. 34). This prompted us to determine whether we could construct satisfactory supramolecular tessellations by using suitable solvent molecules to occupy the cavity and keeping the DNB molecules outside to form exo-complexation.

Previous studies[33] have described that CH$_2$Cl$_2$ could act as a templating solvent for the selective synthesis of P4s because the size of CH$_2$Cl$_2$ molecule fitted well with the cavity size of P4. Besides, P4 could complex one or two CH$_2$Cl$_2$ molecules by multiple C-H⋯π interactions in the solid state. On the basis of that, we assumed that CH$_2$Cl$_2$ molecules would probably take up the cavity of P4, and DNB molecules were expected to be excluded from the cavity, which could possibly result in the exo-complexation between P4 and DNB. Consequently, by slow diffusion of methanol into a solution of P4 and DNB in CH$_2$Cl$_2$, co-crystal of P4-DNB$_b$ crystallized in the triclinic P-1 space group with one P4, two DNB (with each occupancy factor of 0.5), and two CH$_2$Cl$_2$ molecules in the asymmetric unit (Fig. 5c and Supplementary Fig. 36). As expected, with CH$_2$Cl$_2$ molecules occupying the cavity, the DNB molecules were interacted with exo-walls of P4 (Supplementary Fig. 37). As shown in Fig. 5c, the DNB molecules were oriented perpendicularly to the polygon P4 unit plane. Face-to-face π⋯π stacking interactions with distances of 3.46 (A) and 3.39 Å (B) and the corresponding dihedral angles of 10.48° (A) and 1.89° (B) were observed between DNB-3 and DNB-4 with P4. Besides, C-H⋯O interactions (C and D) with distances of 2.64 and 2.68 Å were also observed between the hydrogen atoms of the methoxy groups on P4 and the carbonyl oxygen atoms on two DNB molecules. Moreover, C-H⋯π interaction (E) with the distance of 2.81 Å could also be observed.

Interestingly, two kinds of crystallographically independent DNB molecules (marked as DNB-3 and DNB-4) outside of the P4 walls could work as linkers of four P4 polygon to form the basic parallelogram tiling unit (Fig. 5d, e and Supplementary Figs. 38 and 39). As shown in Fig. 5d, each DNB-3 could be surrounded by four P4 macrocycles and fill the gaps produced by adjacent macrocycles, mainly driven by CT and multiple noncovalent interactions. Among them, DNB-3 could interact with two P4 in a "face-to-face" manner by parallel π⋯π stacking interactions with average plane–plane distances of 3.46 Å (A) and interacted with the other two P4 in a "vertex-to-vertex" manner. Besides, C-H⋯O interactions (B and C) with the distances of 2.68 and 2.59 Å between the oxygen atoms on DNB-3 and the closest methoxy hydrogen atoms of P4 were also observed. Moreover, we found that there existed different noncovalent interactions between adjacent P4 macrocycles (Fig. 5d and Supplementary Fig. 38). When the adjacent P4 were enantiomers with each other, perfect parallel π⋯π stacking interactions with the distances of 3.51 Å (a) and the corresponding dihedral angles of 0° could be observed. For the P4 macrocycles with same configuration, the adjacent P4 interacted with each other by C-H⋯π interactions with the distance of 2.89 Å (b). All these interactions worked together to lead to the stable supramolecular arrangement. Moreover, DNB-4 could interact with four P4 in a similar manner and the detailed interactions (Supplementary Fig. 39) are described in Supplementary Information.

Remarkably, the propagation of these noncovalent interactions in a plane resulted in the formation of a regular superstructure (Fig. 5f). Periodic tiling of P4 with DNB resulted in the formation of a 2D layer-like network superstructure with "non-edge-to-edge" tiling, in which P4 polygon could not meet at edges and were offset with respect to each other (Supplementary Figs. 40 and 41). Further evidence for the fantastic arrangement was demonstrated by the locations of the DNB molecules shown in Fig. 5g, which formed uniform parallelogram grids. Apparently, the arrangement in this crystal structure was totally different from that of the co-crystal of P4-DNB$_a$, demonstrating the significance of solvent modulation in crystallization conditions toward ultimate superstructures. Moreover, we also carried on $^1$H NMR titration experiments of P4 with DNB in both CDCl$_3$ and CD$_2$Cl$_2$ to determine the binding constants (Supplementary Table 1 and Supplementary Figs. 15–18). It was found that the binding constant of P4 with DNB in CDCl$_3$ (622.7 ± 56.8 M$^{-1}$) was much larger than that in CD$_2$Cl$_2$ (96.4 ± 15.0 M$^{-1}$), thus we speculated that the difference in the complexing mode in the solid state was possibly because the size of CH$_2$Cl$_2$ matched the cavity of P4 and competed with DNB, leaving DNB outside of the cavity. In this way, regular supramolecular tessellations could be successfully constructed based on exo-wall interactions between P4 polygon and DNB linkers.

**Supramolecular tessellations of P4 with TPN.** On the basis of the above results, we further showed great interest in whether other guests could be used for regular tessellations. Herein we selected another electron-deficient molecule TPN considering that it has similar planar structure as DNB. First, co-crystal of P4-TPN$_a$ was obtained by slow diffusion of methanol to a solution of P4 and TPN in CHCl$_3$. P4-TPN$_a$ crystallized in the monoclinic P2$_1$/n space group with two P4, two TPN, and two CHCl$_3$ molecules in the asymmetric unit (Fig. 6a and Supplementary Figs. 43 and 44). Similar to P4-DNB$_a$, TPN molecules were encapsulated in the cavities of P4 to form traditional in-cavity host–guest complexation. As shown in Fig. 6b, there existed multiple C-H⋯π interactions between P4-1 and encapsulated TPN-1 with the distances of 2.83 (A), 2.66 (B), 2.84 (C), 2.87 (D), 2.72 (E), 2.74 (F), 2.80 (G), and 2.83 (H) Å, respectively. And there also existed a set of C-H⋯π interactions between P4-2 and encapsulated TPN-2 with the distances of 2.89 (a), 2.77 (b), 2.78 (c), 2.83 (d), 2.63 (e), 2.88 (f), 2.83 (g), 2.80 (h), 2.64 (i), and 2.81 (j) Å, respectively. However, similar to the case of P4-DNB$_a$, the packing of P4 and TPN in P4-TPN$_a$ formed neither tubular superstructure nor a tiling pattern due to the disorderly arranged CHCl$_3$ molecules outside of the cavities (Supplementary Fig. 45).

Co-crystal of P4-TPN$_b$ crystallized in triclinic P-1 space group with one P4, two TPN (with occupancy factor of 0.5), and two CH$_2$Cl$_2$ molecules in the asymmetric unit (Fig. 6c and Supplementary Fig. 47). Similar to the case of P4-DNB$_b$, guest TPN molecules were interacted with exo-walls of P4 since CH$_2$Cl$_2$ molecules occupied in the cavity by a set of Cl⋯π interactions (Supplementary Fig. 48). As shown in Fig. 6c, face-to-face π⋯π interactions with average plane–plane distances of 3.49 (A) and 3.39 Å (B) and the corresponding dihedral angles of 10.45° (A) and 3.63° (B) were observed between TPN-3 and TPN-4 with P4. Besides, C-H⋯π interactions (C) with the distance of 2.82 Å could be observed. Moreover, C-H⋯N interactions with distances of 2.73 (D) and 2.75 (E) were also observed between the hydrogen atoms of P4 and the nitrogen atoms on the cyano groups of TPN.

It was found that the cell parameters and molecular arrangements of P4-TPN$_b$ were almost the same as that of P4-DNB$_b$. Briefly, TPN molecules could be regarded as vertices to link four P4 molecules by CT and multiple non-covalent

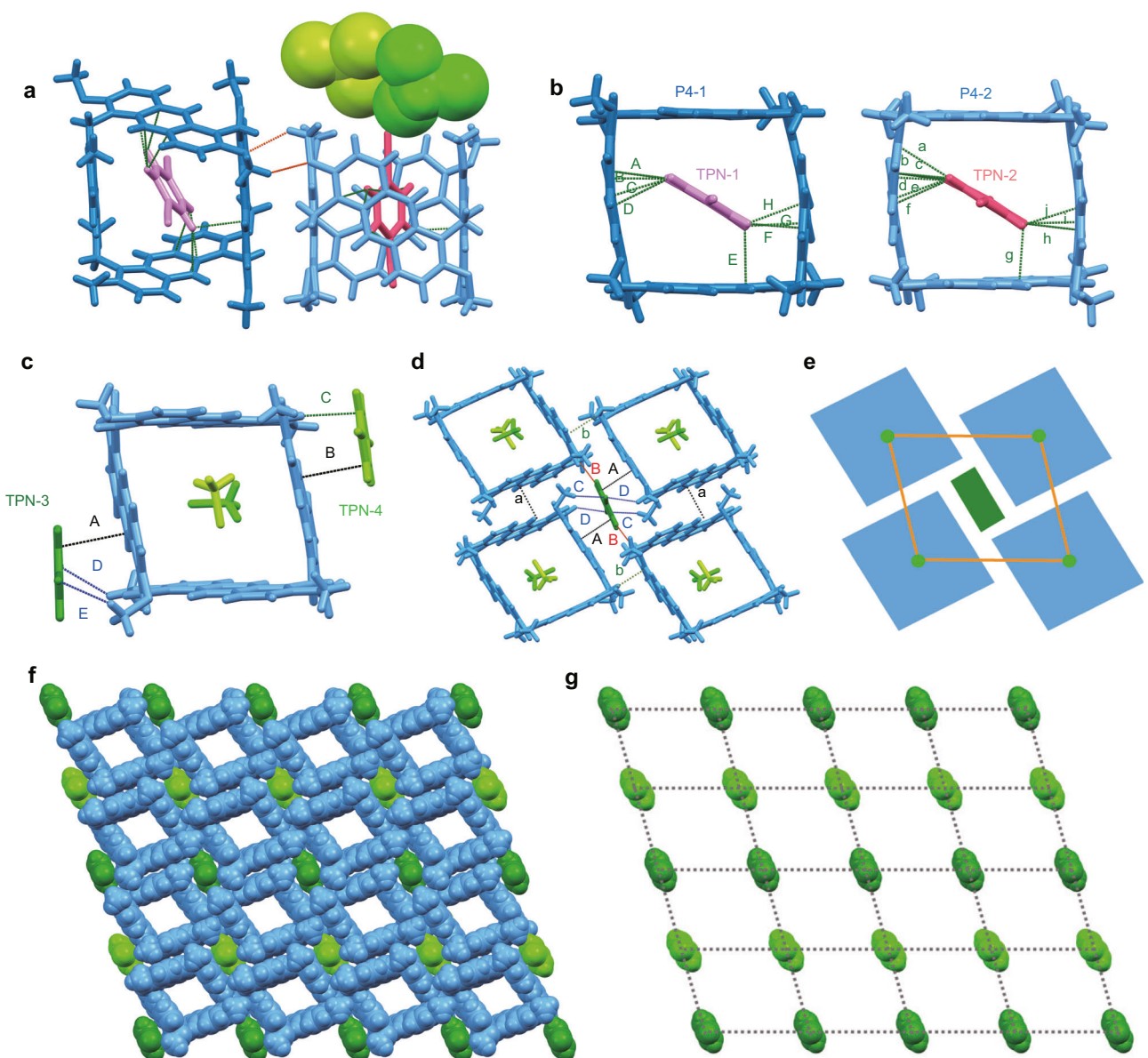

**Fig. 6 Supramolecular tessellations of P4 with TPN.** Crystal structure of P4-TPN$_a$ showing **a** the asymmetric unit; **b** the interactions between P4 and TPN. Crystal structure of P4-TPN$_b$ showing **c** the asymmetric unit; **d** interactions in the basic tiling unit; **e** schematic representation of the basic tiling unit; **f** two-dimensional tiling of the uniform superstructure formed by P4 and TPN in a plane; **g** regular parallelogram grids formed by TPN molecules. Different colors represent the symmetry equivalence. Green, black, and red dotted lines represent the noncovalent interactions. Crystallographically independent P4 and TPN molecules were marked as P4-1, P4-2, TPN-1, TPN-2, TPN-3, and TPN-4, respectively.

interactions as shown in Fig. 6d, e and Supplementary Figs. 49 and 50. TPN-3 exhibited π···π stacking interactions with two P4 with plane–plane distances of 3.49 Å (A), and C-H···O interaction (B) with the distances of 2.67 Å was also observed. In addition, C-H···N interactions (C and D) with distances of 2.73 and 2.75 Å were observed. There also existed different noncovalent interactions between adjacent P4 macrocycles (Fig. 6d and Supplementary Fig. 49). All these interactions worked together to result in the formation of the basic unit of the stable supramolecular arrangements. In common with TPN-3, TPN-4 could interact with four P4 in the same manner and the detailed interactions (Supplementary Fig. 50) are described in Supplementary Information. Remarkably, the whole tiling pattern of P4-TPN$_b$ was also almost the same as that of P4-DNB$_b$. Briefly, TPN molecules worked as vertices and filled the gaps between the four adjacent P4 molecules (Fig. 6f, g). In the whole tiling patterns,

periodic tiling of P4 and TPN also resulted in the formation of a 2D layer-like network superstructure with "non-edge-to-edge" tiling (Supplementary Figs. 51 and 52).

**Supramolecular tessellations of P4 with TFTN in chloroform system.** We further studied the CT assemblies of P4 with guest TFTN, which was more electron-deficient and larger than TPN as hydrogen atoms were replaced with fluorine atoms. Interestingly, unlike the in-cavity complexations formed in both P4-DNB$_a$ and P4-TPN$_a$, P4 and TFTN could form exo-wall complexes in different crystallization conditions including CHCl$_3$ system and exhibit various morphology from macrosystems to microsystems in different systems.

First, red hexagon-shaped co-crystals (Supplementary Fig. 54) of P4-TFTN$_a$ were crystallized by diffusing methanol vapor into a

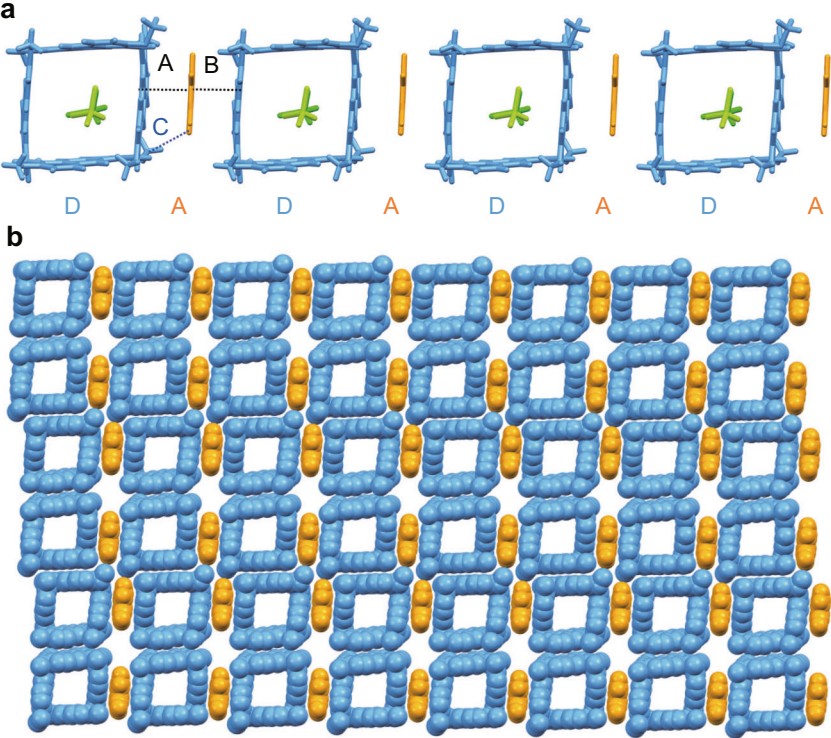

**Fig. 7 Supramolecular tessellations of P4 with TFTN in chloroform system.** Crystal structure of P4-TFTN$_a$ showing **a** the linear DADA-like supramolecular structure in the horizontal direction and **b** 2D layer-like network superstructure. Different colors represent the symmetry equivalence.

solution of P4 and TFTN in CHCl$_3$. P4-TFTN$_a$ crystallized in the triclinic *P*-1 space group with one P4, one TFTN, and two CHCl$_3$ molecules in the asymmetric unit (Fig. 7a and Supplementary Fig. 55). The stoichiometry ratio of P4 with TFTN is 1:1 and TFTN could form exo-complexation with P4 and keep CHCl$_3$ molecules in the cavity probably because TFTN was larger than DNB and TPN and could not fit inside the cavity of P4. As shown in Fig. 7a, each TFTN molecule was sandwiched between two P4 molecules and interacted with the exo-walls of two P4 molecules mainly driven by CT interaction. Multiple noncovalent interactions such as face-to-face π···π stacking interactions (A and B) with the distances of 3.34 and 3.35 Å and corresponding dihedral angles of 2.96° and 2.34°, respectively, and C-H···N interaction (C) with the distance of 2.73 Å and angle of 127.28° were observed. These interactions played an important role in the linear arrangement of P4 and TFTN. As a result, P4 and TFTN formed a one-dimensional DADA-like linear superstructure in the horizontal direction.

It was interesting that we found the linear superstructure could be further cross-linked to a 2D network by π···π stacking interactions between anthracene units of adjacent P4 molecules in the longitudinal direction as shown in Fig. 7b. There existed perfect parallel face-to-face π···π stacking interactions (D and E) with dihedral angles of 0° and plane–plane distances of 3.39 and 3.47 Å. In addition, C-H···O interaction (F and G) with the distance of 2.70 and 2.70 Å were observed (Supplementary Fig. 56). As a result, all these interactions in both the horizontal and the longitudinal directions worked together to result in the formation of 2D layer-like network superstructure.

**Supramolecular tessellations of P4 with TFTN in dichloromethane system.** We further obtained P4-TFTN$_b$, which was crystallized as red rhombic crystals (Supplementary Fig. 58) by diffusing methanol vapor into a solution of P4 and TFTN in CH$_2$Cl$_2$. P4-TFTN$_b$ crystallized in the monoclinic *C2/m* space group with one P4 (occupancy factor of 0.25) and two TFTN as

TFTN-1 and TFTN-2 (occupancy factor of 0.25 and 0.5, respectively) in the asymmetric unit (Supplementary Fig. 59). As shown in Fig. 8a, there were eight TFTN molecules surrounding P4 symmetrically, and all TFTN molecules were oriented perpendicularly to the P4 polygon unit plane. Among them, four TFTN-1 molecules arranged parallel to the anthracene walls of the P4 unit and interacted with the faces of P4 in an extraordinary regular way mainly driven by CT interaction. TFTN-1 molecules established favorable π···π interactions (A) with P4 in face-to-face manner with average distance of 3.31 Å. Meanwhile, four TFTN-2 interacted with four vertices of P4 polygon. Interestingly, two TFTN-2 interacted with P4 in face-to-vertex manner with C-H···N interactions (B) of 2.71 Å and the other two TFTN-2 interacted with P4 in vertex-to-vertex manner through F···O contacts (C) of 2.92 Å. All these interactions worked together to lead to the stable arrangements.

As a result of the propagation of CT interactions in a 2D plane, rhombus-like supramolecular tessellation with 2D layer-like network superstructure was formed by tiling of rhombic P4 polygons with TFTN molecules between two adjacent P4 (Fig. 8b, c). Further evidence for the fantastic arrangement was demonstrated by the location of the TFTN molecules shown in Fig. 8d. Regular rhombic grids were formed by TFTN molecules, which result in a 2D network with a rhombic superstructure. It was found that there existed no interactions between adjacent TFTN molecules indicating TFTN worked as linkers to glue P4 polygons together to form the resulting uniform superstructure. Apparently, the crystal shapes and the tiling patterns in this crystal superstructure were totally different from that of P4-TFTN$_a$ (Supplementary Fig. 60), demonstrating the significance of solvents' modulation in crystallization conditions toward ultimate morphology from macrosystems to microsystems.

**Supramolecular tessellations of P4 with TFTN in acetonitrile system.** Since solvent modulation played a crucial role in the

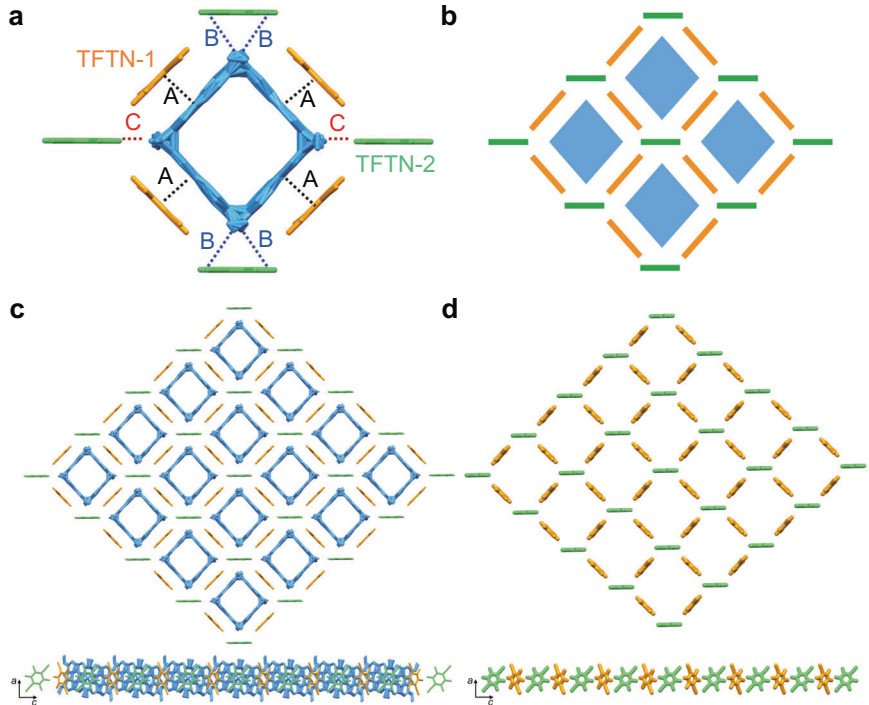

**Fig. 8 Supramolecular tessellations of P4 with TFTN in dichloromethane system.** Crystal structure of P4-TFTN$_b$ showing **a** perfect rhombic P4 with TFTN molecules surrounding each side and vertex of the rhombus unit; **b** schematic representation of the rhombic tiling; **c** two-dimensional regular rhombic tiling formed by P4 and TFTN in a plane; **d** regular rhombic grids formed by TFTN molecules. Different colors represent the symmetry equivalence and hydrogen atoms are omitted for clarity. Crystallographically independent TFTN molecules were marked as TFTN-1 and TFTN-2.

superstructures formed by P4 with TFTN, we further tried other crystallization solvent conditions to see whether we could obtain more supramolecular tessellations based on P4 and TFTN. Luckily, we further obtained P4-TFTN$_c$ as red irregular plates (Supplementary Fig. 62) with brick-like tiling pattern by diffusing methanol vapor into a solution of P4 and TFTN in CH$_3$CN. P4-TFTN$_c$ crystallized in the triclinic *P*-1 space group with one P4, three TFTN marked as TFTN-1, TFTN-2, and TFTN-3 (with each occupancy factor of 0.5), and three CH$_3$CN molecules in the asymmetric unit (Supplementary Fig. 63). As shown in Fig. 9a, three edges of P4 interacted with three TFTN molecules in a face-to-face fashion mainly driven by CT interactions. All the TFTN molecules were oriented perpendicularly to the P4 quadrilateral plane. There existed π···π interactions (A, B, and C) between P4 and three TFTN molecules with the distances of 3.28, 3.27, and 3.29 Å and corresponding dihedral angles of 1.83, 0.75, and 1.67°, respectively. In addition, one P4 could not only encapsulate two CH$_3$CN molecules in the cavity by C-H···π interaction (a) with distance of 2.83 Å but one CH$_3$CN molecule could also be found outside of the cavity and interacted with TFTN-3 by C-H···N interaction (b) with the distance of 2.47 Å.

Interestingly, the other one of the edges of the P4 interacted with adjacent P4 via perfect parallel π···π interactions of 3.47 Å and corresponding dihedral angle of 0° between the anthracene units to form a dimeric basic unit as shown in Fig. 9b and Supplementary Fig. 64. Periodic tiling of the regular parallelogram basic units could result in the formation of a 2D wall-like network superstructure as shown in Fig. 9c and Supplementary Fig. 65. Notably, the TFTN and CH$_3$CN molecules outside the cavity could form regular brick-like grid (Fig. 9d). Especially, CH$_3$CN molecules could be regarded as the vertices of a regular parallelogram and act as linkers to link three TFTN in three perpendicular directions together. From the above results, solvent

modulation played a crucial role in the crystal morphology from macrosystems to microsystems.

## Discussion

In summary, we have demonstrated that P4 composed of four anthracene units to form a square configuration was an excellent macrocyclic building block for the fabrications of various supramolecular tessellations at the molecular level by the exo-wall interactions of the macrocyclic polygons. Consequently, regular rhombic supramolecular tessellations based on P4 itself was constructed by solvent modulation, which could induce an intriguing change in both the shapes and the arrangements of the macrocyclic polygon in crystal superstructures. Notably, the co-crystallization of electron-rich P4 with the electron-deficient guests DNB, TPN, and TFTN through intermolecular CT inter-actions could achieve various highly ordered supramolecular tessellations with different tiling patterns. Interestingly, it was further found that the formation of the resulting co-crystal superstructures exhibited solvent modulation and guest selection. We in this work not only developed a new kind of supramolecular tessellations based on P4 polygon but also provided a new per-spective and opportunity for the design and construction of supramolecular 2D organic materials.

## Methods

**Materials**. All reagents were obtained from commercial sources unless otherwise indicated and used as supplied without further purification: DNB was purchased from Alfa Aesar, TPN was purchased from J&K, and TFTN was purchased from Innochem. The preparation of compound P4[37] is provided in Supplementary Information. The identity and purity of these compounds were confirmed by NMR spectra (Supplementary Figs. 1–4).

**Solution NMR**. [1]H NMR spectra and [19]F NMR spectra were recorded on the Brucker® Avance III 400 MHz NMR spectrometer or Brucker® Avance III 500 MHz NMR spectrometer.

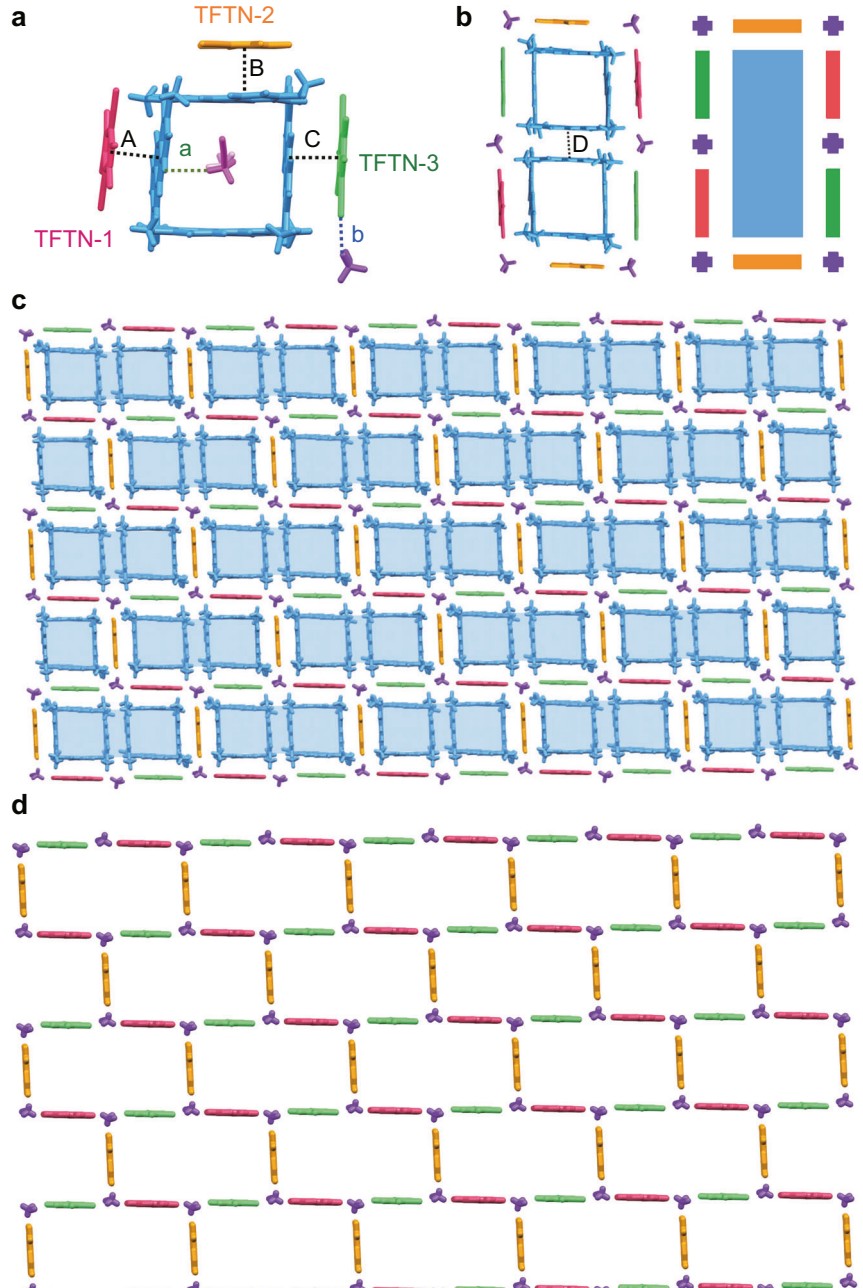

**Fig. 9 Supramolecular tessellations of P4 with TFTN in acetonitrile system.** Crystal structure of P4-TFTN$_c$ showing **a** the asymmetric unit; **b** the basic tiling unit formed by two P4, six TFTN molecules around and six CH$_3$CN molecules as vertexes and its schematic representation; **c** two-dimensional layer-like network superstructure; **d** regular brick-like grids formed by TFTN molecules. Different colors represent the symmetry equivalence and solvent molecules are omitted for clarity. Crystallographically independent TFTN molecules were marked as TFTN-1, TFTN-2, and TFTN-3, respectively.

**UV-vis spectra**. UV-vis absorption spectra were taken on a LAMBDA 950 UV-vis spectrophotometer.

**ESP surface measurements**. All calculations were performed by B3LYP DFT under Gaussian G09. All structures were optimized using 3-21G(d) basis set and then ESP were calculated using 3-21G(d,p) basis set. ESP cubes were generated by Multiwfn code and plotted by the VDW software[62].

**Single crystal X-ray crystallography**. Single crystal X-ray data sets were measured on a Rigaku X-ray diffraction systems XtaLAB Synergy-R. The experimental temperatures were 170 K and the detailed experimental parameters are summarized in Supplementary Tables 2–4. Olex 2 and PLATON software were used for the structure refinement.

**Data availability**

All data supporting this study including detailed methods and experimental details, NMR spectra, UV-vis absorption studies, and crystallographic details of the crystal structures are provided in Supplementary Information. The X-ray crystallographic coordinates for structures reported in this study have been deposited at the Cambridge Crystallographic Data Centre (CCDC), under deposition numbers 2087963, 2087964, 2087965, 2087966, 2087967, 2087969, and 2087970. These data can be obtained free of charge from The Cambridge Crystallographic Data Centre via www.ccdc.cam.ac.uk/data_request/cif. Source data are provided with this paper.

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

## Acknowledgements

We thank the National Natural Science Foundation of China (22031010 and 21521002 to C.-F.C., 91856117 to Y.H.) for financial support. We also thank Mrs. Tongling Liang and Dr. Xiang Hao for their help in the crystal refinements and crystallography descriptions.

## Author contributions

C.-F.C. designed and directed the project. X.-N.H. performed all the experiments and wrote the manuscript. C.-F.C. and X.-N.H. finished the preparation of final manuscript. Y.H. contributed to helpful discussion.

## Competing interests

The authors declare no competing interests.
