## [Peer Review File · Nature Communications]

REVIEWER COMMENTS

Reviewer #1 (Remarks to the Author):

In this manuscript, C. F. Chen et al. constructed a series of tessellation patterns at molecular level by employing their D₄-symmetric pagoda[4]arene (P₄) consisting of anthracene units as tiles. Different solvomorphs, e.g., n-hexane@P₄ and CH₂Cl₂@P₄, were obtained. While the single crystal structure of n-hexane@P₄ does not show a regular 2D pattern, in CH₂Cl₂ P₄ macrocycles form a rhombic tiling pattern on account of the $\pi\cdots\pi$ interactions between the anthracene units of adjacent P₄s. Furthermore, through charge-transfer interactions, electron-rich P₄ was co-crystallised with electron-deficient molecules, e.g., 1,4-dinitrobenzene (DNB), terephthalonitrile (TPN) and tetrafluoroterephthalonitrile (TFTN). Multiple co-crystals with assorted tiling patterns in the crystal structure, depending on the crystallization conditions, were identified.

The act of crystallisation is naturally a tiling process. Molecules consolidate in a periodic manner by virtue of assorted non-covalent interactions to fill the vacancy in space. Therefore, various tiling patterns could be found in any crystal structures, depending on how tiles are defined, as well as the embedded symmetry of both the molecule and the space group. The papers highlighted in Figure 1 by Stoddart and Huang (by the way, the reference numbers 39/40 are not correct) are not that meaningful in terms of crystallography, as 3-, 6-fold symmetric molecules should behave that way. The only novelty there is that the typical Stoddart donor-acceptor continuous stacks were turned into discrete ones spreading in two-dimensions. The reason that those papers could be published in JACS is anyone's guess. A similar paper similar to Stoddart's was recently published in Sci. China. Chem.: see doi.org/10.1007/s11426-021-1030-7.

There is no doubt that C. F. Chen group's pagoda[4]arene is a fantastic macrocycle, and it is indeed a relatively rare 4-fold symmetric molecules with straight walls. It is a piece of solid work with many solvomorphs and polymorphs of pagoda[4]arene, even though the tiling patterns presented are expected. I wonder if the authors have taken a look into A. Szumna and V. Sashuk's pillar[4]pyridinium (see Chem. Commun. 2017, 53, 13320), which is also a square box-like macrocycle. Their paper already featured nice crystal structures with 4-fold tessellation patterns. Similarly, V. Sashuk's pillar[6]pyridinium paper (Chem. Commun. 2018, 54, 6316) also showed very nice honeycomb-like 6-fold tiling. The bottom line is that 3-, 4-, and 6-fold tessellations are not rare in literature and the CCDC database at all. Most authors probably just think it is too trivial to report such patterns.

This is a manuscript which features a lot of beautiful structures and fills in the vacancy of 4-fold tessellation pattern in literature. Honestly, for reasons mentioned above, this probably does not fit into mainstream pure chemistry journals that well but could be very inspiring to the broad readership of Nature Communications. This topic indeed is beyond chemistry and has a lot to do with art and culture. I would suggest the authors to work with someone with crystallography background to use more accurate language and descriptions in the manuscript. In addition, a more

thorough introduction on tessellation from different perspectives and supramolecular tessellation could make the manuscript more presentable to both the general audience and serious chemists. In terms of supramolecular tessellation, the work of N. Champness and P. Beton should not be overlooked. There are also several very nice pieces published by S. De Feyter and Y. Tobe. For 4-fold symmetry, plenty of work was done on porphyrin-based system, no matter on surface or in crystal lattices.

Reviewer #2 (Remarks to the Author):

Tessellations using square molecules reported in this paper would be interesting concept. However, in this paper, the authors report crystal structures of their square-shaped macrocyclic compounds with different guests. Actually, there are many reports of crystal structures of square-shaped molecules such as pillar[4]pyridiniums (Chem. Commun., 2017, 53, 13320) and blue boxes. Their assembled structures also changed depending on kinds of guests. Thus, I'm not sure that the paper is enough novelty to publish in Nat. Commun. About the assemblies of hexagons, I'm not sure Figure 1b is first example or not. Based on pillar[6]arenes, there are many reports of tessellations using their pentagonal structures (For example, Angew. Chem. Int. Ed. 54(22), 6466-6469 (2015), Communications Chemistry 3, 117 (2020) previously.

Reviewer #3 (Remarks to the Author):

Supramolecular Tessellations by the Exo-Wall Interactions of Pagoda[4]arene: Solvent Modulation and Guest Selection

The manuscript is an excellent work done by the Chen's team following their recent seminal work on pagoda[4]arene. The authors came up with a smart way to leverage the unique square shape of pagoda[4]arene in the context of supramolecular tessellations, where they utilized the "exo-wall" interactions between pagoda[4]arene and several electron-deficient planar guests and succeeded in achieving supramolecular tessellations in the solid state by judiciously choosing the solvents used for growing the co-crystals. Although the strategies of both "exo-wall" interactions and solvent modulation are analogous to the prior work reported by Stoddart and Huang separately, the present work complements the two types of 2D tessellation described in Stoddart's and Huang's work, and thus fills the missing link of 2D regular tiling. Moreover, the scholarly presentation of this manuscript is high in that all the figures, particularly those showing the packing modes, are aesthetically pleasing. Therefore, I recommend "minor revision" and would be willing to endorse the acceptance of the manuscript after the authors address the following minor issues properly.

1. The authors suggested that the size of DCM molecule fitted well with the cavity size of P4, thereby allowing the guests to be excluded from the cavity. I am curious if the authors could carry out any titration experiments (e.g., UV-vis titrations) of the P4 host with all three electron-deficient aromatic guests in both DCM and chloroform in an effort to determine the binding constants. If the rationale for solvent modulation is correct, it is expected that P4 would show lower binding affinities with all three electron-deficient aromatic guests in DCM than in chloroform.

2. As opposed to the crystal structures of P4-DNBb and P4-TPNb, no DCM is found inside the cavity of P4 in the crystal structure of P4-TFTNb. The authors should provide a brief rationale for this observation since it is expected that DCM would also occupy the cavity of P4 in the co-crystals of P4 with TFTN grown from the DCM system.

3. Some grammatical errors and formatting issues should be fixed, as detailed below.

Page 3, line 45, Figure 1, the reference numbers for Stoddart's and Huang's work should be 31 and 32, respectively;

Page 4, line 71, change "were" to "was";

Page 5, line 92, change "analysis" to "analyze";

Page 6, line 96, change "form" to "forming";

Page 8, line 144; Page 12, line 206; Page 16, line 286; change "did not form" to "formed";

Page 10, line 173, change "enantiomers alternately arrangement" to "alternately arranged enantiomers"

Page 23, line 411, change "encapsulated" to "encapsulate";

Page 25, line 428, change "could regarded" to "could be regarded";

In SI, the numbering of subheadings for sections 4 and 5 are all mismatched.

Responses to the reviewers' comments:

Thank the reviewers for the valuable comments and suggestions on our manuscript "Supramolecular Tessellations by the Exo-Wall Interactions of Pagoda[4]arene: Solvent Modulation and Guest Selection" (NCOMMS-21-28641), which are very helpful for us to improve our manuscript.

According to the comments and suggestions, the manuscript and SI have been revised, and the point-to-point responses are as follows:

To Reviewer 1:

*In this manuscript, C. F. Chen et al. constructed a series of tessellation patterns at molecular level by employing their D_4 -symmetric pagoda[4]arene (**P4**) consisting of anthracene units as tiles. Different solvomorphs, e.g., n -hexane@**P4** and CH_2Cl_2 @**P4**, were obtained. While the single crystal structure of n -hexane@**P4** does not show a regular 2D pattern, in CH_2Cl_2 @**P4** macrocycles form a rhombic tiling pattern on account of the $\pi\cdots\pi$ interactions between the anthracene units of adjacent **P4**s. Furthermore, through charge-transfer interactions, electron-rich **P4** was co-crystallised with electron-deficient molecules, e.g., 1,4-dinitrobenzene (**DNB**), terephthalonitrile (**TPN**) and tetrafluoroterephthalonitrile (**TFTN**). Multiple co-crystals with assorted tiling patterns in the crystal structure, depending on the crystallization conditions, were identified.*

The act of crystallisation is naturally a tiling process. Molecules consolidate in a periodic manner by virtue of assorted non-covalent interactions to fill the vacancy in space. Therefore, various tiling patterns could be found in any crystal structures, depending on how tiles are defined, as well as the embedded symmetry of both the molecule and the space group. The papers highlighted in Figure 1 by Stoddart and Huang (by the way, the reference numbers 39/40 are not correct) are not that meaningful in terms of crystallography, as 3-, 6-fold symmetric molecules should behave that way. The only novelty there is that the typical Stoddart donor-acceptor continuous stacks were turned into discrete ones spreading in two-dimensions. The reason that those papers could be published in JACS is anyone's guess. A similar paper

similar to Stoddart's was recently published in *Sci. China. Chem.*: see doi.org/10.1007/s11426-021-1030-7.

Q1. There is no doubt that C. F. Chen group's pagoda[4]arene is a fantastic macrocycle, and it is indeed a relatively rare 4-fold symmetric molecules with straight walls. It is a piece of solid work with many solvomorphs and polymorphs of pagoda[4]arene, even though the tiling patterns presented are expected. I wonder if the authors have taken a look into A. Szumna and V. Sashuk's pillar[4]pyridinium (see *Chem. Commun.* 2017, 53, 13320), which is also a square box-like macrocycle. Their paper already featured nice crystal structures with 4-fold tessellation patterns. Similarly, V. Sashuk's pillar[6]pyridinium paper (*Chem. Commun.* 2018, 54, 6316) also showed very nice honeycomb-like 6-fold tiling. The bottom line is that 3-, 4-, and 6-fold tessellations are not rare in literature and the CCDC database at all. Most authors probably just think it is too trivial to report such patterns.

Answer: Thank the reviewer very much for the comment. It is the truth that A. Szumna and V. Sashuk's pillar[4]pyridinium (*Chem. Commun.* **2017**, 53, 13320) is also a square macrocycle and it can show a nice crystal superstructure with 4-fold tessellation pattern. However, compared with that case of pillar[4]pyridinium, our work have demonstrated that pagoda[4]arene (**P4**) with a square configuration can not only form nice crystal superstructures itself, but also show interesting solvent-controllable behaviors in both shapes and superstructures. **More importantly and meaningfully, it was found that the macrocycle can be utilized as an excellent macrocyclic building block for fabrication of various supramolecular tessellations through exo-wall host-guest interactions.** Consequently, in addition to the 4-fold tessellation with rhombic tiling pattern by **P4** itself through favorable $\pi \cdots \pi$ interactions between the anthracene units of adjacent **P4**. Notably, various highly ordered supramolecular tessellations with different tiling patterns were fabricated by the rare explored exo-wall charge-transfer interactions between electron-rich **P4** (donor) and electron-deficient guests (acceptors) under the different crystallization conditions. Very interestingly, solvent modulation and guest selection played a vital role in controlling the molecular arrangements of the

resulting crystal superstructures. In a word, **our work proves that pagoda[4]arene is an excellent square building block for the fabrication of various supramolecular tessellations, and the utilization of exo-wall host-guest interactions also provides a new perspective and opportunity for design and construction of supramolecular two-dimensional organic materials.** Thus, we believe that our work is significant and novelty enough to be published in *Nature Communications*.

Q2. This is a manuscript which features a lot of beautiful structures and fills in the vacancy of 4-fold tessellation pattern in literature. Honestly, for reasons mentioned above, this probably does not fit into mainstream pure chemistry journals that well but could be very inspiring to the broad readership of Nature Communications. This topic indeed is beyond chemistry and has a lot to do with art and culture. I would suggest the authors to work with someone with crystallography background to use more accurate language and descriptions in the manuscript.

Answer: Thank the reviewer for the comments and suggestions. According to the suggestions, we have consulted to Dr. Xiang Hao with crystallography background in our Institute, and revised our manuscript under his help. Detailed changes have been highlighted with yellow color in the manuscript (see Page 11, line 11; Page 13, line 9; Page 13, line 22; Page 15, line 19; Page 17, line 7; Page 19, line 7; Page 20, line 19; Page 23, line 2).

Q3. In addition, a more thorough introduction on tessellation from different perspectives and supramolecular tessellation could make the manuscript more presentable to both the general audience and serious chemists.

Answer: According to the suggestion of the reviewer, we have added a more thorough introduction on tessellation and supramolecular tessellation (see Page 2, line 2 and line 11) in the manuscript.

Q4. In terms of supramolecular tessellation, the work of N. Champness and P. Beton should not be overlooked. There are also several very nice pieces published by S. De Feyter and Y. Tobe. For 4-fold symmetry, plenty of work was done on porphyrin-based system, no matter on surface or in crystal lattices.

Answer: According to the suggestion of the reviewer, we have cited the representative works related to supramolecular tessellation of N. Champness and P. Beton (see ref. 23-25) and S. De Feyter and Y. Tobe (see ref. 26).

To Reviewer 2:

Q. Tessellations using square molecules reported in this paper would be interesting concept. However, in this paper, the authors report crystal structures of their square-shaped macrocyclic compounds with different guests. Actually, there are many reports of crystal structures of square-shaped molecules such as pillar[4]pyridiniums (Chem. Commun., 2017, 53, 13320) and blue boxes. Their assembled structures also changed depending on kinds of guests. Thus, I'm not sure that the paper is enough novelty to publish in Nat. Commun. About the assemblies of hexagons, I'm not sure Figure 1b is first example or not. Based on pillar[6]arenes, there are many reports of tessellations using their pentagonal structures (For example, Angew. Chem. Int. Ed. 54(22), 6466-6469 (2015), Communications Chemistry 3, 117 (2020) previously.

Answer: Thank the reviewer for the comments. It is true that pillar[4]pyridiniums with square-shape can show nice assembled structures. However, compared with that case of pillar[4]pyridinium, our work have demonstrated that pagoda[4]arene (**P4**) with a square configuration can not only form nice crystal superstructures itself, but also show interesting solvent-controllable behaviors in both shapes and superstructures. More importantly and meaningfully, it was found that the macrocycle can be utilized as an excellent macrocyclic building block for fabrication of various supramolecular tessellations through exo-wall host-guest interactions. Consequently, in addition to the 4-fold tessellation with rhombic tiling pattern by **P4** itself through favorable $\pi \cdots \pi$ interactions between the anthracene units of adjacent **P4**. Notably, various highly ordered supramolecular tessellations with different tiling patterns were fabricated by the rare explored exo-wall charge transfer interactions between electron-rich **P4** (donor) and electron-deficient guests (acceptors) under the different crystallization conditions. Very interestingly, solvent modulation and guest selection played a vital role in controlling the molecular arrangements of the resulting crystal superstructures. In a

word, our work proves that pagoda[4]arene is an excellent square building block for the fabrication of various supramolecular tessellations, especially, the utilization of the exo-wall host-guest interactions also provides a new perspective and opportunity for the design and construction of supramolecular two-dimensional organic materials. Thus, we believe that our work is significant and novelty enough to be published in *Nature Communications*.

In addition, although there are reports of pillar[6]arene-based macrocycles with pentagonal structures that showed nice hexagon assemblies, as far as we know, Huang's work is the first to utilize pillar[6]arene as a hexagon building block to construct hexagon supramolecular tessellations via exo-wall interactions of pillar[6]arene and electron-deficient guests. Thus, we highlighted Huang's work in Figure 1b as a representative hexagon supramolecular tessellation building block.

To Reviewer 3:

The manuscript is an excellent work done by the Chen's team following their recent seminal work on pagoda[4]arene. The authors came up with a smart way to leverage the unique square shape of pagoda[4]arene in the context of supramolecular tessellations, where they utilized the "exo-wall" interactions between pagoda[4]arene and several electron-deficient planar guests and succeeded in achieving supramolecular tessellations in the solid state by judiciously choosing the solvents used for growing the co-crystals. Although the strategies of both "exo-wall" interactions and solvent modulation are analogous to the prior work reported by Stoddart and Huang separately, the present work complements the two types of 2D tessellation described in Stoddart's and Huang's work, and thus fills the missing link of 2D regular tiling. Moreover, the scholarly presentation of this manuscript is high in that all the figures, particularly those showing the packing modes, are aesthetically pleasing. Therefore, I recommend "minor revision" and would be willing to endorse the acceptance of the manuscript after the authors address the following minor issues properly.

Q1. *The authors suggested that the size of DCM molecule fitted well with the cavity size of P4, thereby allowing the guests to be excluded from the cavity. I am curious if the*

authors could carry out any titration experiments (e.g., UV-vis titrations) of the P4 host with all three electron-deficient aromatic guests in both DCM and chloroform in an effort to determine the binding constants. If the rationale for solvent modulation is correct, it is expected that P4 would show lower binding affinities with all three electron-deficient aromatic guests in DCM than in chloroform.

Answer: Thank the reviewer for the comments and suggestions. According to the suggestions of the reviewer, we have carried on ^1H NMR titration experiments of **P4** with the three related electron-deficient guests in both CDCl_3 and CD_2Cl_2 to determine the binding constants. Related discussion was added in the manuscript (see Page 15, line 6) and supplementary information (Table S1 and Fig. S13-Fig. S21). As expected, it was found that for guests **DNB** and **TPN**, both the binding constants of **P4** with them were much larger in CDCl_3 than in CD_2Cl_2 probably because the size of solvent CD_2Cl_2 matched the cavity of **P4** and competed with guests **DNB** and **TPN**, thus indicating the solvent modulation was reasonable. For guest **TFTN**, the binding constants in both CDCl_3 and CD_2Cl_2 were too small to be accurately calculated probably because **TFTN** was larger than **DNB** and **TPN**, and it could not fit inside the cavity of **P4**. These results were consistent with the case that **P4** and **TFTN** could form exo-wall complexations in both chloroform and DCM system in the solid states.

***Q2.** As opposed to the crystal structures of P4-DNB_b and P4-TPN_b, no DCM is found inside the cavity of P4 in the crystal structure of P4-TFTN_b. The authors should provide a brief rationale for this observation since it is expected that DCM would also occupy the cavity of P4 in the co-crystals of P4 with TFTN grown from the DCM system.*

Answer: In the crystal structure of P4-TFTN_b, there is high probability of solvent molecules inside the cavity of **P4** and most likely to be DCM since the crystal of P4-TFTN_b was grown from DCM system. However, the solvent molecules were severely disordered and could not to be identified accurately, thus we imposed SQUEEZE treatment with PLATON during the refinement.

***Q3.** Some grammatical errors and formatting issues should be fixed, as detailed below. Page 3, line 45, Figure 1, the reference numbers for Stoddart's and Huang's work*

should be 31 and 32, respectively;

Page 4, line 71, change “were” to “was”;

Page 5, line 92, change “analysis” to “analyze”;

Page 6, line 96, change “form” to “forming”;

Page 8, line 144; Page 12, line 206; Page 16, line 286; change “did not form” to “formed”;

Page 10, line 173, change “enantiomers alternately arrangement” to “alternately arranged enantiomers”

Page 23, line 411, change “encapsulated” to “encapsulate”;

Page 25, line 428, change “could regarded” to “could be regarded”;

In SI, the numbering of subheadings for sections 4 and 5 are all mismatched.

Answer: Thank the reviewer very much for the suggestions, which are very helpful for improving our manuscript. According to the suggestion of reviewer, we have adjusted the reference numbers for Stoddart’s and Huang’s work to be 35 and 36, respectively (Page 3, Fig. 1). And revised all the grammatical errors and formatting issues as suggested by the reviewer (Page 4, line 16 : “were” has been changed to “was”; Page 6, line 1 : “analysis” has been changed to “analyze”; Page 6, line 5 : “form” has been changed to “forming”; Page 8, line 16; Page 12, line 2; Page 16, line 7: “did not form” have been changed to “formed”; Page 10, line 13: “enantiomers alternately arrangement” has been changed to “alternately arranged enantiomers”; Page 23, line 10, “encapsulated” has been changed to “encapsulate”; Page 25, line 8: “could regarded” has been changed to “could be regarded”. In SI, the numbering of subheadings for sections 4 and 5 have been revised.

REVIEWERS' COMMENTS

Reviewer #1 (Remarks to the Author):

The authors have already addressed the issues raised in my previous report. Overall this manuscript is in a good shape.

Reviewer #3 (Remarks to the Author):

The authors have addressed all the concerns.

We are grateful to the reviewers for the comments on our manuscript “Supramolecular Tessellations by the Exo-Wall Interactions of Pagoda[4]arene: Solvent Modulation and Guest Selection” (NCOMMS-21-28641A), which are very helpful for us to improve our manuscript.

Reviewer #1 (Remarks to the Author): The authors have already addressed the issues raised in my previous report. Overall this manuscript is in a good shape.

Response: Thank the reviewer very much for the comments on our manuscript.

Reviewer #3 (Remarks to the Author): The authors have addressed all the concerns.

Response: Thank the reviewer very much for the comments on our manuscript.